# Chondrocyte Spheroids Laden in GelMA/HAMA Hybrid Hydrogel for Tissue-Engineered Cartilage with Enhanced Proliferation, Better Phenotype Maintenance, and Natural Morphological Structure

**DOI:** 10.3390/gels7040247

**Published:** 2021-12-02

**Authors:** Guanhuier Wang, Yang An, Xinling Zhang, Pengbing Ding, Hongsen Bi, Zhenmin Zhao

**Affiliations:** Department of Plastic Surgery, Peking University 3rd Hospital, Beijing 100191, China; jasperwang_0624@163.com (G.W.); anyangdoctor@163.com (Y.A.); xinlingpku@163.com (X.Z.); dingpengbing917@163.com (P.D.)

**Keywords:** bioengineered cartilage, cross-linkable hybrid hydrogel, spheroid

## Abstract

Three-dimensional cell-laden tissue engineering has become an extensive research direction. This study aimed to evaluate whether chondrocyte spheroids (chondro-spheroids) prepared using the hanging-drop method could develop better cell proliferation and morphology maintenance characteristics, and be optimized as a micro unit for cartilage tissue engineering. Chondro-spheroids were loaded into a cross-linkable hybrid hydrogel of gelatin methacrylate (GelMA) and hyaluronic acid methacrylate (HAMA) in vivo and in vitro. Cell proliferation, aggregation, cell morphology maintenance as well as cartilage-related gene expression and matrix secretion in vitro and in vivo were evaluated. The results indicated that compared with chondrocyte-laden hydrogel, chondro-spheroid-laden hydrogel enhanced proliferation, had better phenotype maintenance, and a more natural morphological structure, which made it appropriate for use as a micro unit in cartilage tissue engineering.

## 1. Introduction

Nasal and auricular reconstruction and esthetic surgery have high strength and shape requirements for augmentation materials; hence, cartilage has been widely applied for contour-filling. At present, the commonly used autogenous cartilage for augmentation includes the conchal cartilage, the costal cartilage, and the septal cartilage [1]. The acquisition of cartilage is associated with pain, surgical wounds, and risk of complications [2]. In addition, the cartilage needs to be cut and carved for contouring, which further increases donor site loss. In cases of nasal reconstruction and secondary repair of rhinoplasty, there is a further shortage of autologous cartilage materials [3]. Tissue-engineered cartilage has provided a new method for the repair of cartilage defects and improvement of esthetic contour.

To achieve these surgical goals, cell-laden scaffolds, which include hydrogels, polymeric hard materials, and hydrophilic glass fibers, have been widely investigated [4,5,6,7]. In recent years, the influence of biomechanics on three-dimensional (3D) culture has been a popular research topic. In the case of chondrogenic differentiation of stem cells, studies have shown that the mechanical properties of hydrogels affect the chondrogenic fate of stem cells in 3D culture [8]. In a low-stress environment, MSCs tend to differentiate into adipocytes; and undergo osteogenic differentiation in a high-stress environment [9,10]. Moreover, studies have shown that the rigidity, micromorphology, and printing graphics of hydrogels affect chondrogenic outcome [11]. Therefore, selecting a hydrogel scaffold with proper mechanical strength is necessary to optimize the bioengineered cartilage.

To date, many types of hydrogel materials have been used as cell-laden scaffolds, such as collagen [12,13], sodium alginate [14], hyaluronic acid (HA) [7,15], and gelatin [16]. The advantages of methacrylated gelatin (GelMA) include better solubility and lower immunogenicity [17]. GelMA exhibits a more positive effect on the 3D culture of mesenchymal stem cells (MSCs) and chondrocytes. Many studies have confirmed that GelMA can support extracellular matrix formation as well as chondrocyte phenotype maintenance [18,19]. Methacrylated hyaluronic acid (HAMA) has greater mechanical strength than GelMA. HAMA has been proven to be suitable for chondrocyte culture in recent studies [17]. A hybrid gel composite of GelMA and HAMA can enhance the strength of materials while ensuring extracellular matrix formation and chondrocyte phenotype maintenance, and was proven to be more conducive to chondrogenic differentiation [10,20].

In 3D culture scaffolds, the final products are closely related to properties of cells, cell growth environment, scaffold material properties, biomechanical environment, and chemical addition factors [11,21]. Therefore, the choice and forms of seed cells are also important.

Seed cells commonly applied in cartilage tissue engineering include chondrocytes, chondrocyte progenitor cells, and mesenchymal stem cells (MSCs) [22]. Hydrogels containing various concentrations of seed cells laden in various biomaterials have been widely studied [18,22]. These isolated seed cells were able to proliferate, secrete extracellular matrix, and form bioengineered cartilage. However, they could not satisfactorily simulate the living environment of chondrocytes; hence, the phenotype of chondrocytes was not finely maintained, and the histological structure was quite different from that of natural cartilage. Scaffold-free 3D culture techniques to produce cell spheroids [23], such as hanging-drop microplates, magnetic levitation, and spheroid microplates with ultra-low attachment coating may provide a new way to solve above problems. Spheroids are cell aggregates that can self-assemble in an environment that prevents attachment to a flat surface [24]. The hanging-drop method is a relatively easy method to obtain spheroids. In addition to their application in drug testing [25,26] and nanoparticle examination [27,28] and being used as a micro model to study diseases [29,30], spheroids are also used for regenerative medicine in the form of injectable suspensions [31,32,33] and bio-printed constructs [34].

Research has shown that compared to injection of cells alone, MSC spheroids laden in fibrin hydrogel had enhanced proangiogenic and anti-inflammatory potential, and result in better wound healing outcome by localizing cells at the defect site and upregulating trophic factor secretion [35]. In addition, MSC spheroids were reported to promote neuronal differentiation [36], and MSC-derived Schwann cell spheroids can promote peripheral nerve regeneration [37]. Some studies have been conducted on the application of spheroids in cartilage bioengineering. Moor et al. [6] bio-printed MSC spheroids with GelMA hydrogel in 3D lattice constructs, and estimated that MSC spheroids can be bio-printed as microtissue while maintaining cell viability, 3D architecture, chondrogenic phenotype, and fusion capacity.

In this study, chondrocyte spheroids (chondro-spheroids) prepared using the hanging-drop method were loaded into cross-linkable gelatin methacrylate (GelMA)/hyaluronic acid methacrylate (HAMA) hybrid hydrogels [17,38], which proved to be suitable for chondrocytes. We aimed to evaluate whether chondro-spheroids could perform better cell proliferation and morphology maintenance, produce more abundant cartilage extracellular matrix while forming bionic histological structures closer to natural cartilage, and whether they could be used as a micro unit for 3D bio-printing.

## 2. Results and Discussion

### 2.1. Characterization of the Hydrogel Constructs

The lyophilized GelMA and HAMA powders were processed and the procedure is shown in Figure 1. In brief, 12.5% GelMA, 2% HAMA, and 8%/0.8% GelMA/HAMA solutions were prepared. After being crosslinked with blue light, the 12.5% GelMA, 2% HAMA, and 8% GelMA/0.8% HAMA hybrid hydrogel constructs were obtained.

The porosity of each hydrogel construct was 69%, 85%, and 81%, respectively. The SEM image of the surface of the 8%/0.8% GelMA/HAMA hybrid hydrogel construct is shown in Figure 1E.

The rheological properties of the 12.5% GelMA, 2% HAMA, and 8% GelMA/0.8% HAMA hybrid hydrogels were analyzed as a function of temperature, as shown in Figure 1F. The viscosity curve of HAMA changes linearly with a small range. The viscosity is 100–150 mPa/s at room temperature (5–35 °C). The viscosity curve of GelMA shows a significant difference in the room temperature range. When it is less than 20 °C, the viscosity increases rapidly to 10^4^ mPa/s, which makes it more difficult to produce constructs during the experiment. The hybrid hydrogel balances the rheological properties of HAMA and GelMA gels; it displays greater viscosity, which is not as affected by temperature, making the production conditions more stable.

A compression test was performed to identify the difference in mechanical properties. The hybrid hydrogel constructs had an average compression modulus of 38.3 kPa, which is between GelMA (25.1 kPa) and HAMA (39.6 kPa), as shown in Figure 1G.

### 2.2. Viability and Stability of Spheroid during the Hanging-Drop Process

#### 2.2.1. Formation of Chondro-Spheroids Using the Hanging-Drop Culture Method

When chondrocytes reached 80–90% confluence at passage 2, the monolayer was enzymatically dissociated with 0.25% trypsin. The cell suspension was counted using a hemocytometer and diluted to 3.5 × 10^5^ cells/mL. A volume of 35 μL of cell solution per drop (containing approximately 12,000 cells) was prepared and placed onto the cover of the culture plate. Cell drops were cultured for 48 h in an incubator at 37 °C with 5% CO_2_. The chondro-spheroids were collected using an Eppendorf pipette and centrifuged at 800 rpm for 3 min. A schematic diagram of the chondrocyte spheroid production process is shown in Figure 2A,D–G.

#### 2.2.2. Viability and Stability of Chondrocyte Spheroid

At 6, 24, 48 and 72 h after self-constructing in the hanging drop, chondro-spheroids were collected and observed, as shown in Figure 3A. In the first 48 h of construction, the diameter of spheroids gradually shrinks and the spheroids gradually compact. At 6 h, cells agglomerate in a flaky fashion. After centrifugation and resuspension, the morphology changed significantly. There were approximately 12,000 cells in each hanging drop, and the average diameter was 502 ± 86 µm (average ± standard deviation). After 24 h, a quasi spheroid formed, the shape remained unchanged after centrifugation and resuspension, and the diameter was 312 ± 52 µm. After 48 h, a round spheroid formed, the shape remained unchanged after centrifugation and resuspension, and the diameter decreased to 267 ± 18 µm. In the first 48 h, the living cells rate remained above 95%. At 72 h, the spheroid stopped shrinking while the living cells rate decreased significantly to 72%. Therefore, it was judged that 48 h for hanging-drop culture is the duration to prepare chondrocyte spheroids. The percentage of live cells in spheroids is shown in Figure 3D, and the change in diameter of spheroids over 72 h is shown in Figure 3E.

### 2.3. The Spheroid-Laden Construct and the Cell-Laden Construct In Vitro

#### 2.3.1. Formation of the Spheroid-Laden Construct and the Cell-Laden Construct

The spheroid-laden constructs were made with a density of 400 spheroids/200 μL of hybrid hydrogel. Additionally, the cell-laden constructs were made with a density of 5 × 10^6^ cells /200 μL of hybrid hydrogel. A schematic diagram of the production process of spheroid-laden constructs is shown in Figure 2B, and a schematic diagram of the production process of cell-laden constructs is shown in Figure 2C. The surface morphology and structure of laden cells and laden spheroids were examined by scanning electron microscopy (SEM) and are shown in Figure 3(C1–C4).

#### 2.3.2. Optical Microscopy Performance and Live/Dead Staining of Spheroid-Laden Construct and Cell-Laden Construct Samples In Vitro

These in vitro observations lasted for 7 days. Optical microscopy revealed that the cells in the outer layer of chondro-spheroids began to spread on day 1 after embedding the hydrogel. The spheroid diameter gradually increased from 268 ± 20 µm in day 1 to 422 ± 46 µm in day 7 as the cells began to proliferate. In suspended chondrocytes in the CON group, the cells began to spread in the gel on day 1, and most of them remained spherical-like in the 3D space. These results are shown in Figure 3B. In the live/dead assay, which was observed through confocal microscopy, most of the chondrocytes were alive at 0, 3, and 7 days in both the EXP and CON groups. The percentage of live cells in the spheroid-laden construct and the cell-laden construct is shown in Figure 3F, and the change in diameter of spheroids in 7 days is shown in Figure 3G.

### 2.4. The Spheroid-Laden Construct and the Cell-Laden Construct In Vivo

#### 2.4.1. Gross Observation

The gross views of the constructs before and after implantation of 1 and 2 months are shown in Figure 4A–F. The changes in the diameter and wet weight of the constructs are shown in Figure 4G,H. The diameter of the CON group significantly increased after 1 month of implantation, which changed from transparent to translucent. After 2 months, the texture of the CON group was uneven, and some calcification points developed (as shown in Figure 4F). In the EXP group, the diameter of the constructs was slightly reduced; however, the diameter of the cartilage spheroids in the constructs increased and fusion between spheroids occurred. After 2 months, the module changed from translucent to almost opaque. Wide fusion occurred between the spheres (as shown in Figure 5 by the black arrow).

We found that chondro-spheroids maintained good viability in hybrid hydrogels and a spherical shape while proliferating by 2 fold in diameter. In addition, neighboring spheroids were observed to be connected and merged. These characteristics make it suitable for use as a micro unit for bio-printing to create a bioengineered cartilage module.

#### 2.4.2. Histological Evaluation of Spheroid and Cell Samples In Vivo

The HE staining, safranin O/fast green staining, and toluidine blue staining of the EXP and CON groups at 1 and 2 months, and the diced natural auricular cartilage in hybrid hydrogel (N-C) are shown in Figure 5. Clear nuclei and obvious cell structures were observed in all sections. Cartilage lacunae were formed in both groups. However, in the EXP group, the distribution of the extracellular matrix and histological morphology of lacunae in the chondro-spheroids were very similar to those of natural cartilage (i.e., the positive control). The average diameter of the chondro-spheroid was 266 ± 13 μm before implantation, 335 ± 62 μm after 1 month, and 508 ± 147 μm after 2 months, as shown in Figure 4I.

As shown with safranin O/fast green staining and toluidine blue staining, proliferated cells and increased extracellular matrix were observed in chondro-spheroids after 1 month. The cartilage lacunae at the center of the spheroids were relatively round and appeared deep blue/red with special staining. The cartilage lacunae at the lateral portion of the spheroids were relatively narrow and appeared light blue/red with special staining, indicating that the extracellular matrix in the lateral portion has not yet been fully secreted. After 2 months, the diameter of the cell spheroids doubled, and the secretion of the extracellular matrix was close to the natural cartilage. Mature lacunae were also observed in the connection between spheroids, and some spheroids merged into a whole. In the CON group, suspended chondrocytes were evenly distributed in the hybrid gel, and glycosaminoglycan secretion was observed after 1 month. The cartilage lacunae were well formed and enlarged. The extracellular matrix was mainly deposited in the limited area around the lacunae and did not match the histological morphology of the natural cartilage.

#### 2.4.3. IHC Evaluation of Spheroid and Cell Samples In Vivo

The immunohistochemical staining outcomes of the EXP, CON, and N-C groups are shown in Figure 6A. After 1 and 2 months, significantly more intensive COL II staining was observed in the EXP group than in the CON group and the N-C group. Meanwhile, significantly positive COL X staining was observed in the EXP group compared with the CON group, indicating that fewer chondrocytes were de-differentiated. Significantly more intensive Ki-67 was observed at 1 than at 2 months in both groups, indicating that chondrocyte proliferation was active at 1 month and tended to be stable at 2 months. Interestingly, Ki-67 was observed in most chondrocytes of the spheroids at 1 month, and it was only observed in the outer layer of the spheroids at 2 months. The AOD and APSAP evaluation outcomes are shown in Figure 6B,C, respectively.

The immunohistochemical staining outcome proved that chondro-spheroids could have more extracellular matrix secretion, and were less hypertrophic. A significant sign of proliferation was observed 1 month after in vivo implantation, and proliferation activity occurred on the outer layer of the spheroids after 2 months, which was consistent with the characteristics of cell spheroid proliferation reported previously [39].

#### 2.4.4. Gene Expression

The gene expression levels of the EXP group, the CON group, and the natural cartilage group were analyzed in Figure 7. The gene expression outcome of the EXP group and the CON group before embedding, at 1 month and 2 months after embedding, are shown in Figure 7A–D. Before embedding, the expression levels of Sox9, Col10A1 and HIF-1a in the EXP group and the CON group were similar. The expression of Col2A1 in the EXP group was slightly higher, which was considered to be caused by culturing in hanging drops for 48 h. Compared with the CON group, the expression of chondrogenic gene marker Col2A1 (* *p* < 0.05 in 1 month, ** *p* < 0.01 in 2 months), transcription factor Sox9 (* *p* < 0.05 in 1 month, ** *p* < 0.01 in 2 months) and hypoxia-inducible factor HIF-1a (* *p* < 0.05 in 1 month and 2 months) in the EXP group was significantly upregulated. The expression of the Col10A1 gene in the CON group was significantly higher than that in the EXP group (** *p* < 0.05 in 1 month). After 2 months of in vivo implantation, the expression of Col2A1, Sox9 and HIF-1a in the EXP group was higher than that in natural cartilage, and the expression of Col10A1 was similar to that in natural cartilage.

The gene expression of the EXP and CON groups at 2 months was also compared with the gene expression of natural cartilage. The expression of Col2A1, Sox 9, Col10A1 and HIF-1a in natural cartilage was used as reference, and the relative expression in the EXP and CON groups were recorded, as shown in Figure 7E.

The RT-qPCR outcome was consistent with histological and IHC staining that the expression of Col2A1, Sox9 of the chondro-spheroid were upregulated and Col10A1 was downregulated. Interestingly, gene expression of HIF-1a was significantly higher in chondro-spheroid, which hinted that chondrocytes in spheroid had better phenotype maintenance [40] compared with suspended chondrocytes. The local activation of the hypoxia-inducible transcription factor HIF-1α may have resulted in the hypoxic environment of the chondro-spheroid since it was dense.

In conclusion, these results show that the EXP group can promote gene expression related to chondrocyte differentiation and maintain chondrocyte phenotype by inhibiting de-differentiation and hypertrophy, and can achieve the goal of tissue engineering construction closer to natural cartilage.

## 3. Conclusions

In this study, chondro-spheroids prepared using the hanging-drop method were loaded into the GelMA/HAMA hybrid hydrogel constructs. Compared with normal chondrocyte-laden hydrogel constructs through in vitro and in vivo experiments, the chondro-spheroids were shown to have better performance in terms of cell proliferation and aggregation, and maintain chondrocyte phenotype by inhibiting de-differentiation and hypertrophy, and to produce abundant cartilage extracellular matrix, while forming bionic histological structures closer to natural cartilage.

## 4. Materials and Methods

### 4.1. Hydrogel Preparation and Characterizations

#### 4.1.1. Preparation of the Hybrid Gel

The lyophilized GelMA and HAMA powders were processed as previously described [41]. Briefly, a solution of porcine skin gelatin type A in phosphate buffer was prepared. Methacrylic anhydride was then added dropwise into the gelatin solution. The mixture was diluted, filtered and dialyzed against distilled water for 1 week and lyophilized (Figure 1A). A solution of hyaluronic acid in distilled water was prepared. Methacrylic anhydride was added dropwise while stirring. After adjusting the pH, it was incubated overnight, dialyzed against distilled water for 3 days and lyophilized (Figure 1B). The lyophilized GelMA powder was dissolved in DMEM at 80 °C, and the 12.5% gelatin solution was harvested. The lyophilized HAMA powder was dissolved in DMEM at 37 °C, and the 2% hyaluronic acid solution was harvested. Similarly, both GelMA and HAMA powders were dissolved in DMEM, and the 8%/0.8% GelMA/HAMA solution was harvested. All solutions were sterilized after filtration through a 0.22 μm strainer (Falcon). To construct the hybrid hydrogel, the GelMA/HAMA solution and photoinitiator LAP (2.5% w/v) were mixed in a ratio of 10:1. The solution was exposed to 405 nm blue light for 30 s to form the hydrogel construct (Figure 1C,D).

#### 4.1.2. Rheological Properties of Hydrogels

The 10% GelMA, 2% HAMA, and 8%/0.8% GelMA/HAMA solutions were prepared as described above. LAP was added at a concentration of 0.25% w/v. Rheological evaluation was performed as previously described [5]. Briefly, on a MCR rheometer (AntonPaar, Graz, Austria), the shear stress and viscosity were measured at each temperature while the gels were subjected to a temperature decrease ramp in the range of 37–15 °C with a decrease rate of 5 °C/min at a constant shear rate of 100 (1/s).

#### 4.1.3. Mechanical Testing

Compression testing of photo-crosslinked hydrogel composed of 10% GelMA, 2% HAMA, and 8%/0.8% GelMA/HAMA was conducted as previously described. Briefly, on the samples using TA HD plus texture analyzer (Stable micro systems), each specimen was compressed at a displacement rate of 1 mm/min to a maximum force of 5 kg. All tests were performed in triplicates. Young’s modulus (MPa) was calculated from the stress–strain curve as the slope of the initial linear portion of the curve (5–10%), with any toe region due to the initial settling of the specimen neglected.

#### 4.1.4. Porosity Analysis by SEM

After photocrosslinking, the 12.5% GelMA, 2% HAMA, and 8%/0.8% GelMA/HAMA hydrogel constructs were frozen in liquid nitrogen, lyophilized and sputter coated with Au-Pd (2 nm). The constructs were examined with field emissions scanning electron microscope (JSM-7900F JEOL Ltd., Tokyo, Japan). Porosity was calculated with Image J software. Three samples and 3 images from each sample were taken in each hydrogel.

### 4.2. Cell Harvest and Culture

Ear cartilage was obtained from euthanized New Zealand rabbits (male, 2–2.5 kg, n = 4), according to the guidelines of the Institutional Animal Care and Use Committee. Under aseptic conditions, the skin and the intact perichondrium were carefully removed, and the remaining ear cartilage tissues were cut into small pieces (1–2 mm^2^). The diced cartilage was washed with calcium- and magnesium-free PBS (pH 7.4) and digested for 12 h at 37 °C with 0.2% type IV collagenase (Sigma-Aldrich, Shanghai, China) in PBS. After filtration through a 70 μm strainer (Falcon), the cells were washed with PBS and collected by centrifugation (1000 rpm, 5 min). The collected cells were cultivated in DMEM/F12 containing 15% fetal bovine serum and 1% antibiotic-antimycotic solution (Thermo Fisher Scientific, Waltham, MA, USA) at 37 °C with 5% CO_2_, and the culture medium was changed every 3 days.

### 4.3. Chondro-Spheroid Formation and In Vitro Characterization

#### 4.3.1. Formation of Chondro-Spheroids Using the Hanging-Drop Culture Method

When chondrocytes reached 80–90% confluence at passage 2, the monolayer was enzymatically dissociated with 0.25% trypsin. Cell suspension was counted using a hemocytometer and diluted to 3.5 × 10^5^ cells/mL. A volume of 35 μL of cell solution per drop (containing approximately 12,000 cells) was prepared and placed onto the cover of the culture plate. Cell drops were cultured for 48 h in an incubator at 37 °C with 5% CO_2_. The chondro-spheroids were collected using an Eppendorf pipette and centrifuged at 800 rpm for 3 min. A schematic diagram of the chondrocyte spheroid production process is shown in Figure 2A,D–G.

#### 4.3.2. Viability and Stability of Chondrocyte Spheroid

Following the above protocol, the chondro-spheroids self-constructed in the hanging drop for a total of 72 h. At 6, 24, 48 and 72 h, 10 chondro-spheroids were collected, respectively. The morphology of spheroids was observed under an optical microscope, and the diameter was measured. The spheroids were then centrifuged (1000 rpm, 5 min) and resuspended 30 times, and the morphology was observed again. Another 10 chondro-spheroids were collected for live/dead staining. Fluorescein diacetate (FDA, 1 μg/mL) and propidium iodide (PI, 1 μg/mL) were used, and viability was observed by confocal laser scanning microscopy (Leica STED, Wetzlar and Mannheim, Germany). The number of living cells were counted.

#### 4.3.3. Preparation of the Spheroid-Laden Gel and the Cell-Laden Gel

Centrifugated chondro-spheroids were collected and mixed with GelMA and HAMA solutions. Then, an appropriate volume of DMEM was added to adjust the final concentration of the gel to 8%/0.8% (GelMA/HAMA). In the group of spheroids (experimental group, EXP group), a photoinitiator (LAP, 0.25% w/v in final) was added, and the mixture of spheroids and hybrid hydrogel was piped into the culture dish, with a density of 400 spheroids/200 μL of hybrid hydrogel. In the group of suspended separated cells (control group, CON group), LAP was added, and the mixture of chondrocytes and hybrid hydrogel was piped into the culture dish at a density of 5 × 10^6^ cells/200 μL of hybrid hydrogel. The solution was exposed to 405 nm blue light for 30 s. Then, the cell-laden and spheroid-laden hydrogel constructs were washed with PBS.

#### 4.3.4. Characteristics of Spheroid and Cell Samples In Vitro

All samples were cultivated in DMEM containing 15% fetal bovine serum and 1% antibiotic-antimycotic solution at 37 °C with 5% CO_2_, and the culture medium was changed every other day. An optical microscope was used to observe the morphological changes in spheroids and cells in vitro. After the samples were cultured for 1, 3, and 7 days in vitro, they were stained with fluorescein diacetate (FDA, 1 μg/mL) and propidium iodide (PI, 1 μg/mL), and the viability and distribution were observed by confocal laser scanning microscopy (Leica STED, Wetzlar and Mannheim, Germany).

### 4.4. In Vivo Experiments

#### 4.4.1. Establishment of Animal Model

Samples for in vivo experiments were implanted in nude mice as soon as they were prepared, as described previously. This animal study was approved by the Institutional Animal Care and Use Committee, and all procedures were strictly performed following the guidance of care and use of laboratory animals. Twelve male BABL/c nude mice (6–8 weeks old, approximately 20 g) were used. After intraperitoneal injection with 10 mg/mL pentobarbital sodium, a 1 cm skin incision was made on each dorsal site, and two subcutaneous pockets for disc implantation were prepared under aseptic and anesthetic conditions. Twelve hydrogel discs of chondrocyte cells (CON group) were placed on the left side. Twelve hydrogel discs of chondro-spheroids (EXP group) were placed on the right side. The skin was closed using interrupted suture. Collection of the chondrocyte and all the animal experiments were approved by the Ethics Committee of Peking University Health Science Center (LA2021374).

#### 4.4.2. Gross Observation and Biomechanical Evaluation of Samples In Vivo

Before implantation, the diameters of 24 constructs in the two groups were accurately recorded with a Vernier caliper, and the constructs were weighed (wet weight). At 1 and 2 months after implantation, the nude mice were euthanized, six structures were removed from each group, the diameter of the structures were measured using a Vernier caliper, and then the structures were weighed (wet weight).

#### 4.4.3. Histological and Immunohistochemical Staining of Spheroid and Cell Samples In Vivo

After harvesting at 1 and 2 months, three samples in each group were fixed in 4% paraformaldehyde solution at 4 °C overnight. To further compare the engineered cartilage tissues with natural cartilage, fresh diced rabbit ear cartilage fragments (approximately 1 mm^2^) were also embedded in the hybrid hydrogel and fixed in 4% paraformaldehyde for staining tests. For histological staining, samples were dehydrated in graded alcohol concentrations and embedded in paraffin. Paraffin sections of 5 μm thickness were cut, deparaffinized, and rehydrated. Sections were stained with hematoxylin/eosin (HE), toluidine blue, and safranin O/fast green staining, in accordance with standard protocols. The diameter of each spheroid was measured using the NDP view 2 software (Hamamatsu, Japan) and recorded. For immunohistochemical staining, the fixed samples were embedded in OCT and sectioned at 5 μm. Collagen II (1:200, Novus), collagen X (1:50, Abcam), and Ki-67 (1:100, Novus) were detected using mouse anti-rabbit collagen II primary antibody, collagen X primary antibody, and Ki-67 primary antibody, respectively. After performing procedures in accordance with standard protocols, positive immunohistochemical staining was analyzed quantitatively using IPP 6.0, and the value of the average optical density (AOD) and average positive staining area percentage (APSAP) was measured for use as the evaluation index.

#### 4.4.4. RNA Analysis In Vivo

Three samples at each group were collected in liquid nitrogen for RNA analysis. Total RNA was extracted with TRIzol (Invitrogen) and RNeasy Mini Kit (QIAGEN). The cDNA was synthesized from 2 μg total RNA by cDNA Reverse Transcription Kit (Invitrogen). Quantitative reverse transcription PCR (RT-qPCR) was performed using the SYBR Green PCR Master Mix (Applied Biosystem, Foster City, CA, USA) and the ABI ViiA 7 Real Time PCR System (Applied Biosystems) following a thermal profile: 2 min at 95 °C, and 40 cycles each of 10 s at 94 °C, 10 s at 59 °C, and 40 s at 72 °C. The relative mRNA levels were calculated using the ΔΔCt method and Gapdh was used as reference gene in each reaction. As for the relative gene expression of the EXP and CON groups compared with the natural cartilage, the expression of Col2A1, Sox 9, Col10A1 and HIF-1a in natural cartilage was used as reference. The primers used are listed in Table 1

### 4.5. Statistical Analysis

All data are presented as the mean ± standard deviation (SD). Statistical analysis was performed using Student’s *t*-test. Differences between groups were considered statistically significant at * *p* < 0.05, ** *p* < 0.01, and *** *p* < 0.001.

## Figures and Tables

**Figure 1 gels-07-00247-f001:**
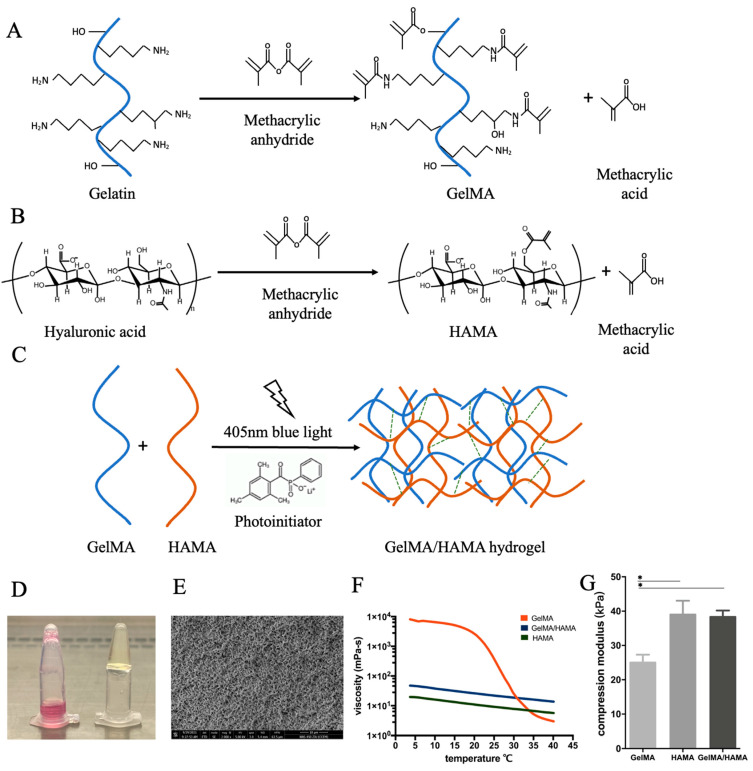
Assembly of the hydrogel construct from GelMA and HAMA polymer chains. Prior to gel assembly, gelatin and hyaluronic acid are methacrylated separately and lyophilized for storage. (**A**) The binding of methacrylate groups to the primary NH2 groups of gelatin. (**B**) The binding of methacrylate groups to the hydroxyl groups of hyaluronic acid. (**C**) To create a hybrid hydrogel, a solution of methacrylated gelatin and methacrylated hyaluronic acid was formed by rehydrating lyophilized powders of both, then crosslinked by the addition of a photoinitiator (LAP) and 405 nm blue light irradiation for 30 s. (**D**) GelMA/HAMA hydrogel solution before (left) and after (right) 30 s of blue light treatment. (**E**) SEM of the cross-section of the GelMA/HAMA hydrogel (scale bar: 10 µm). (**F**) Rheological properties of the 12.5% GelMA, 2% HAMA, and 8% GelMA/0.8% HAMA hydrogels. Viscosity of three different solutions as a function of temperature was measured at a shear rate of 100 s^−1^. (**G**) Young’s modulus of the 12.5% GelMA, 2% HAMA, and 8% GelMA/ 0.8% HAMA hydrogel constructs (* *p* < 0.05).

**Figure 2 gels-07-00247-f002:**
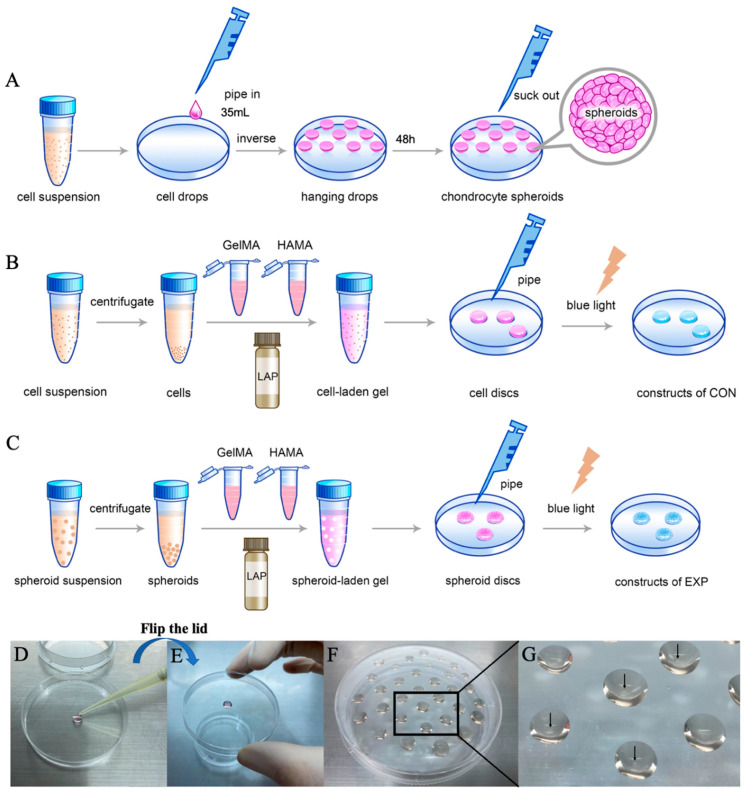
The schematic diagram of the production processes of chondro-spheroids (**A**), cell-laden hydrogel constructs (**B**), and spheroid-laden hydrogel constructs (**C**). During the procedure, a 200 µL pipette gun is used to extrude a 35 µL cell suspension on the Petri dish lid (**D**), and then the lid is quickly turned over by hand to prepare the hanging drop (**E**). Multiple hanging drops can be prepared on one lid and taken out after 48 h (**F**). It can be seen by naked eye that the chondrocyte spheroids are formed in hanging drops as shown by the black arrows (**G**).

**Figure 3 gels-07-00247-f003:**
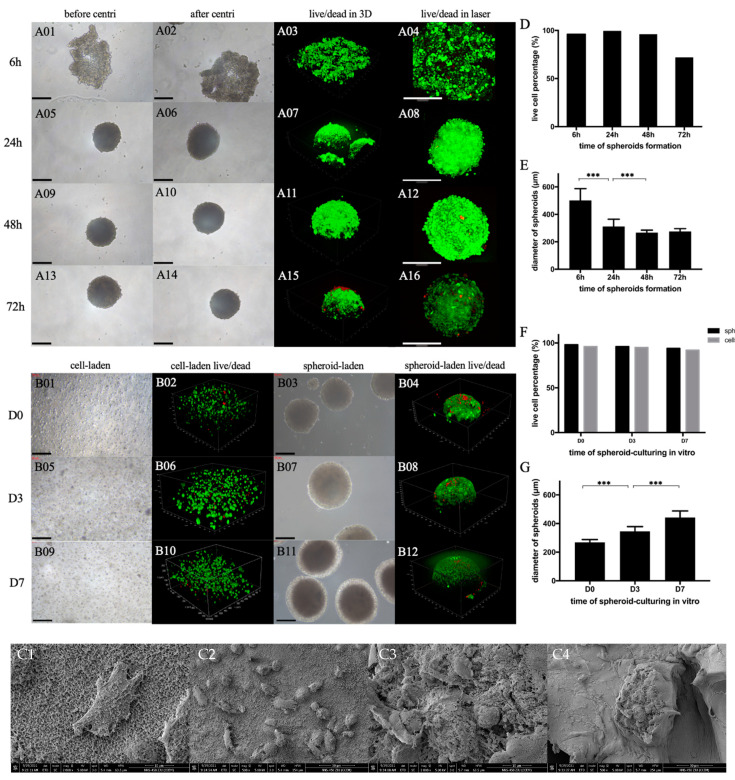
(**A**) The process of spheroid formation in the hanging drops over 72 h. The spheroid morphology before centrifugation, after centrifugation, and the live/dead staining outcome in 3D view and vertical view at 6 h (**A01**–**A04**), 24 h (**A05**–**A08**), 48 h (**A09**–**A12**), and 72 h (**A13**–**A16**), respectively. Inverted microscopy: 10×, confocal microscopy: 25×. Scale bar = 200 μm. (**B**) In vitro culturing of spheroid-laden hydrogel constructs and cell-laden hydrogel constructs. The cell morphology, the live/dead staining outcome of cells, the spheroid morphology, the live/dead staining outcome of spheroids at day 1 (**B01**–**B04**), day 3 (**B05**–**B08**), and day 7 (**B09**–**B12**), respectively. Inverted microscopy: 10×. Confocal microscopy: 25×. Scale bar = 200 μm. (**C**) The SEM images of cell-laden hydrogel constructs (**C1**,**C2**) and spheroid-laden hydrogel constructs (**C3**,**C4**). Scale bar as shown in figures. Dispersed chondrocytes adhered to the hydrogel. Chondrocytes in spheroid are closely connected with each other. (**D**) The percentage of live cells at different time periods during spheroid formation in hanging drops. (**E**) The diameter changes in spheroids during spheroid formation in hanging drops. (**F**) The percentage of live cells at different time periods during the in vitro culturing of hydrogel constructs. (**G**) The diameter changes in spheroids during the in vitro culturing of hydrogel constructs. (*** *p* < 0.001.)

**Figure 4 gels-07-00247-f004:**
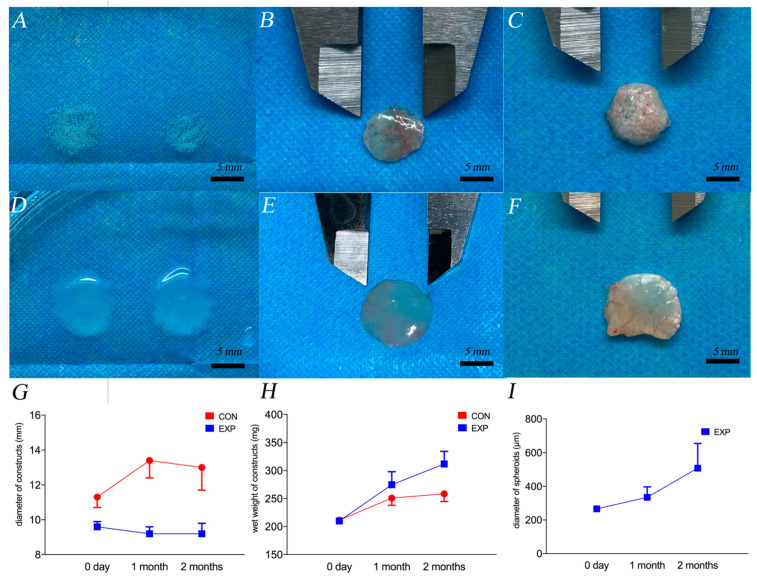
The gross view of constructs in the EXP group before implantation (**A**), 1 month after implantation (**B**), and 2 months after implantation (**C**). The gross view of constructs in the CON group before implantation (**D**), 1 month after implantation (**E**), and 2 months after implantation (**F**). Scar bar = 5 mm. The calcification points in the CON group at 2 months are marked as red arrows. (**G**) The changes in the diameter of constructs, (**H**) the wet weight of constructs, and (**I**) the diameter of chondro-spheroids. The blue spots and lines represent the EXP group, and the red spots and lines represent the CON group.

**Figure 5 gels-07-00247-f005:**
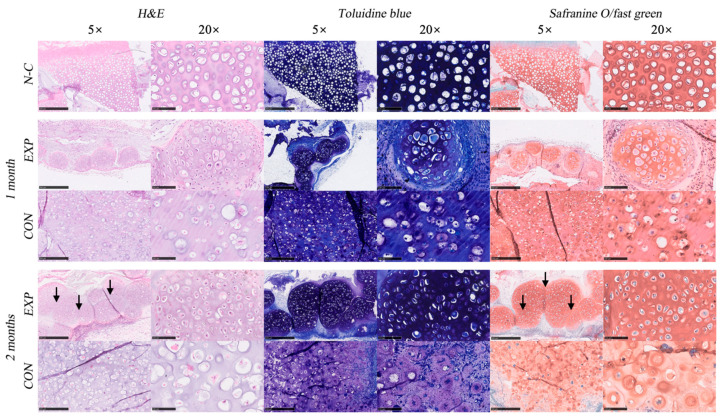
The H&E staining, toluidine staining, and safranin O/fast green staining of natural cartilage (N-C), the EXP group, and the CON group, after 1 month and 2 months of implantation, under the microscope with a magnification of 5× (scale bar = 500 μm) and 20× (scale bar = 100 μm). From the H&E staining and safranin O/fast green staining of the EXP group at 2 months, we can observe the fusion and merge of neighboring spheroids, as shown by the black arrows.

**Figure 6 gels-07-00247-f006:**
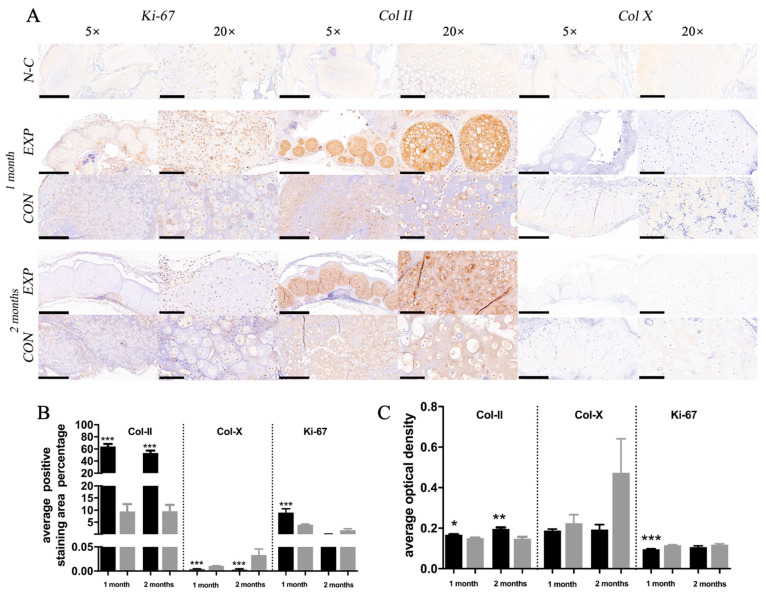
The immunohistochemical staining outcome including Ki-67, Col II, and Col X of natural cartilage (N-C), the EXP group and the CON group, after 1 and 2 months of implantation, under the microscope with a magnification of 5× (scale bar = 500 μm) and 20× (scale bar = 100 μm) (**A**). The AOD (**B**) and APSAP (**C**) evaluation outcomes of the immunohistochemical staining including Col-II, Col-X, and Ki-67 in the EXP and CON groups (* *p* < 0.05, ** *p* < 0.01, *** *p* < 0.001).

**Figure 7 gels-07-00247-f007:**
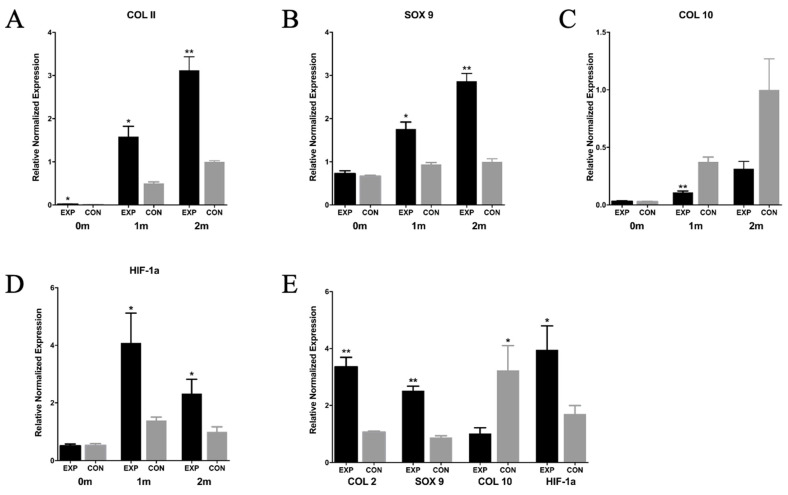
The quantitative gene expression by real-time PCR of the EXP group, the CON group and the N-C group in vivo. (**A**–**D**) The expression of Col2A1, Sox-9, Col10A1, and HIF-1a of EXP group and CON group before implantation, 1 month and 2 months after implantation. Gapdh was used as reference gene. (**E**) The expression of Col2A1, Sox-9, Col10A1, and HIF-1a of EXP group and CON group at 2 months after implantation, in which the expression of Col2A1, Sox 9, Col10A1 and HIF-1a in natural cartilage was used as reference, respectively. (n = 3, * *p* < 0.05, ** *p* < 0.01.)

**Table 1 gels-07-00247-t001:** Primers used in the RT-qPCR.

Gene	Forward Primer	Reverse Primer
Col2A1	GGCCTACCTGGATGAAGCCA	GGTGAACCTGCTGTTGCCCT
Sox9	AGTACCCGCACCTGCACAAC	CGCTTCTCGCTCTCGTTCAG
Col10A1	ATGCCCGATGACTTTACAAA	AGGCATTCCTGTTACTCCCT
HIF-1a	ACAGTGTTCCCGCAGACTCA	TTTAATCGTCAGTGGTGGCG
Gapdh	CACGGTCAAGGCTGAGAACG	TCACCCCACTTGATGTTGGC

## Data Availability

Data are contained within the article.

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
