# Peer review of "Chondrocyte Spheroids Laden in GelMA/HAMA Hybrid Hydrogel for Tissue-Engineered Cartilage with Enhanced Proliferation, Better Phenotype Maintenance, and Natural Morphological Structure"

_gels, 2021, doi:10.3390/gels7040247_

Round 1

Reviewer 1 Report

In this paper, the authors prepared chondrocyte spheroids-laden hydrogels for engineered cartilage, which presents enhanced proliferation, better phenotype maintenance and more natural morphological structure. However, much of the discussion is misleading and important aspects of the analysis methods and results are missing. The author should state the results more clearly and accurately. In addition, extra attention should be paid to spelling and grammar. There are many remaining issues to be resolved before consideration of publication.

In the following, there is a list of questions that the authors should answer:

1) Page 3, lines 110-112: Regrading “In brief, 10% GelMA, 2% HAMA, and 8%/0.8% GelMA/HAMA solutions were prepared. After crosslinked with blue light, the 12.5% GelMA, 2% HAMA, and 8% GelMA / 0.8% HAMA hybrid hydrogel construct were obtained.” Why the concentration of GelMA is increased from 10% to 12% after crosslinking. The groups of 12.5% GelMA, 2% HAMA, and 8%/0.8% GelMA/HAMA are not reasonable for comparison. It’s better to use the groups of 8% GelMA, 0.8% HAMA, and 8%/0.8% GelMA/HAMA.

2) In Fig.1G, Fig 3D~G, the error bars and significant difference analysis are missing. In Fig 3A and B, the scale bars are missing.

3) Page 7, line 208: The calcification points is not clear in the images, it should be marked.

4) Page 8, lines 214 and 215: The sentence “In addition, neighboring spheroids were observed to be connected and merged” could not be concluded from the images in Fig.4. The evidence should be provided to demonstrate how you reached your conclusion.

5) Cartilage is avascular, but blood vessels are visible in Fig.4B,F, please explain this issue.

6) In Fig.6B,C, the natural cartilage group should be included as a positive control.

7) Page 11, lines 279 and 289: The sentence “the expression of Col10A1 was similar to that in natural cartilage” cannot be concluded, because the author did not investigate the gene expression of natural cartilage which should be included as a positive control.

8) In Materials and Methods, the methods that investigate the porosity and SEM images of hydrogels should be described.

9)The engineered cartilage prepared by chondrocyte spheroids-laden hydrogels is not homogeneous, since the size of neighboring spheroids was uneven as well their gaps were observed even after 2 months of implantation, which may cause the mechanical instability of implanted tissues. Please discuss this issue.

Author Response

Reply letter to reviewer:

Dear reviewer,

Thank you very much for reviewing this article.

Let me answer your question as follows.

1) Page 3, lines 110-112: Regrading “In brief, 10% GelMA, 2% HAMA, and 8%/0.8% GelMA/HAMA solutions were prepared. After crosslinked with blue light, the 12.5% GelMA, 2% HAMA, and 8% GelMA / 0.8% HAMA hybrid hydrogel construct were obtained.” Why the concentration of GelMA is increased from 10% to 12% after crosslinking. The groups of 12.5% GelMA, 2% HAMA, and 8%/0.8% GelMA/HAMA are not reasonable for comparison. It’s better to use the groups of 8% GelMA, 0.8% HAMA, and 8%/0.8% GelMA/HAMA.

Answer:

Sorry for the mistake that 12.5% was written as 10%. We have corrected this mistake. We used 2.5% GelMA and 2% HAMA as a control because these two concentrations of gels were reported for cartilage tissue engineering and proved to be suitable for cartilage regeneration. The concentrations of 8% GelMA and 0.8% HAMA were too small for cartilage regeneration.

2) In Fig.1G, Fig 3D~G, the error bars and significant difference analysis are missing. In Fig 3A and B, the scale bars are missing.

Answer: Thank you for your advice. These significant difference analysis in Fig. 1G was added. And the scale bars in Fig 3A and B were marked in a more obvious bar. To be noted, the scale bar in confocal 3D images were included in the axis. No other scale bar was added in the confocal 3D images. The figure legend of Fig 1 and 3 are also corrected.

As for Fig 3D-G, the significant difference analysis in fig E and G were added, and relative figure legends are also corrected. Figure D and F are the live cell percentage of spheroids. This data is calculated by computer estimation, which is not accurate. So, we suggest that the average data is enough to indicate the degree of living cells.

3) Page 7, line 208: The calcification points are not clear in the images, it should be marked.

Answer: Thank you for your advice. We have added the arrow and correct the figure legend.

4) Page 8, lines 214 and 215: The sentence “In addition, neighboring spheroids were observed to be connected and merged” could not be concluded from the images in Fig.4. The evidence should be provided to demonstrate how you reached your conclusion.

Answer: Thank you for your advice! The neighboring spheroids were observed to be connected and merged through histological staining. We apologize the misunderstanding caused by the lack of expression. We have revised relative sentences in lines 214 and 215. And we added arrows to show the merge in figure 5.

5) Cartilage is avascular, but blood vessels are visible in Fig.4B,F, please explain this issue.

Answer: dear reviewer, thank you for your help. The blood vessels in fig 4B and F are located outside of the hydrogel, in the capsule between the subcutaneous tissue and the hydrogel, they do not enter the cartilage tissue, which can be seen from histological staining.

6) In Fig.6B,C, the natural cartilage group should be included as a positive control.

Answer: dear reviewer, thank you for your help. We added the words “positive control” in the manuscript. In the figure, the abbreviation as “N-C” might be fine, since lots of published articles have used the “N-C” to represent for natural cartilage, and it is easy to understand quickly.

7) Page 11, lines 279 and 289: The sentence “the expression of Col10A1 was similar to that in natural cartilage” cannot be concluded, because the author did not investigate the gene expression of natural cartilage which should be included as a positive control.

Answer: dear reviewer, the gene expression of natural cartilage indeed has been processed in our research, as showed in figure 7E. We apologize that the manuscript did not express our mind clearly. We have added the relative text in “2.4.4. The gene expression” and “4.4.4. RNA analysis in vivo”.

8) In Materials and Methods, the methods that investigate the porosity and SEM images of hydrogels should be described.

Answer: dear reviewer, thank you for your help. We added the part “4.1.4. Porosity analysis by SEM” in the manuscript.

9)The engineered cartilage prepared by chondrocyte spheroids-laden hydrogels is not homogeneous, since the size of neighboring spheroids was uneven as well their gaps were observed even after 2 months of implantation, which may cause the mechanical instability of implanted tissues. Please discuss this issue.

Answer: dear reviewer, thank you for your help. In this paper, we mainly testified that chondrocyte spheroids as a micro unit had better cartilage properties in hybrid hydrogels. As for the creation of more homogeneous materials, they can be obtained by 3D printing, and we will continue to deepen this research.

Reviewer 2 Report

The manuscript titled "Chondrocyte spheroids laden in GelMA/HAMA hybrid hydrogel for tissue-engineered cartilage with enhanced proliferation, better phenotype maintenance, and natural morphological structure" by Wang et al. is a well-written manuscript which will be of interest to all researchers in the field of cartilage tissue engineering.

Major comments:

The manuscript will benefit if the following comments are addressed-
1. Please provide the full forms of any acronym used at the first instance (e.g. Line  - HA, GelMA etc.)
2. Figure 2: Please check if 1B and 1C are labeled correctly. It seems 1B is cell laden hydrogels and 1C is spheroid laden (the figure legend labels them oppositely at present).

3. Figure 3: 
(a) Please include the time-points in the image itself. Please also clarify whether A01, A05, A09 and A13 are before centrifugation and A02, A06, A10 and A14 are after centrifugation (similar query for the live/ dead assay).
(b) Why are there no error bars in 3D and 3F. The x-axis of 3F should be re-labeled since it also shows cell laden cultures as well
(c) Comment on the morphology observed in SEM images
4. Figure 4:
(a) Please mention the sample types on top of the images itself - it will be easy for readers.
(b) Comment on Figure 4F: Is there vascularization observed in cell-laden hydrogels?
4. Mention the ethics number and date.

Author Response

Reply letter to reviewer 1:

Dear reviewer,

Thank you very much for reviewing this article.

Let me answer your question as follows.

  1. Please provide the full forms of any acronym used at the first instance (e.g. Line - HA, GelMA etc.)

Answer: Dear reviewer, thank you for your help. We have revised and corrected these acronyms.

  1. Figure 2: Please check if 1B and 1C are labeled correctly. It seems 1B is cell laden hydrogels and 1C is spheroid laden (the figure legend labels them oppositely at present).

Answer: Dear reviewer, thank you for your help. We feel sorry to make such a mistake. It has been corrected.

  1. Figure 3:

(a) Please include the time-points in the image itself. Please also clarify whether A01, A05, A09 and A13 are before centrifugation and A02, A06, A10 and A14 are after centrifugation (similar query for the live/ dead assay).

Answer: Dear reviewer, thank you for your help. We added the time-points in row and the image category in columne, so that each image is clarified.

(b) Why are there no error bars in 3D and 3F. The x-axis of 3F should be re-labeled since it also shows cell laden cultures as well

Answer: Dear reviewer, thank you for your help. Figure 3D and 3F are the live cell percentage of spheroids. This data is calculated by computer estimation, which is not accurate. So, we suggest that the average data is enough to indicate the degree of living cells. As for figure 3F, we used the side boxes (Black: spheroids, gray: cells).

(c) Comment on the morphology observed in SEM images

Answer: Dear reviewer, thank you for your help. We have added comment on the morphology observed in SEM images.

  1. Figure 4:

(a) Please mention the sample types on top of the images itself - it will be easy for readers.

Answer: Dear reviewer, thank you for your help. We have added sample typeson top of the images.

(b) Comment on Figure 4F: Is there vascularization observed in cell-laden hydrogels?

Answer: dear reviewer, thank you for your help. The blood vessels in fig 4B and F are located outside of the hydrogel, in the capsule between the subcutaneous tissue and the hydrogel, they do not enter the cartilage tissue, which can be seen from histological staining.

  1. Mention the ethics number and date.

Answer: dear reviewer, thank you for your help. We have added the ethics number and date in the manuscript.

Reviewer 3 Report

The article is written in a very interesting way. The reviewed work will certainly interest the readers of the magazine. I recommend the publication after a few minor additions: - what is the thermal effect of cross-linking polymers used in the research? is there an increase in temperature in this process? - have the authors studied the gel content of the prepared scaffolds? This parametr is a very important parameter to assess the colonization of cells on the scaffold. After these doubts are rectified, I recommend the publication. 

Author Response

Reply letter to reviewer 3:

Dear reviewer,

Thank you very much for reviewing this article.

Let me answer your question as follows.

What is the thermal effect of cross-linking polymers used in the research? is there an increase in temperature in this process? - have the authors studied the gel content of the prepared scaffolds?

Answer: Dear reviewer, thank you very much for your help with the review process, and thank you a lot for your encouragement. However, we failed to find any reports about the thermal effects of GelMA and HAMA hydrogel in other studies. In practice, we did not feel any special thermal effect of GelMA and HAMA hydrogel on cells. The gel content ratio has been systematically studied in other articles. In this paper, only one suitable choice is used to prove the advantage of cartilage spheroids. I'm sorry for not answering your question in detail. I hope to get your understanding.

Round 2

Reviewer 1 Report

The authors have carefully revised the manuscript according to the reviewer’s comments, and the quality is greatly improved. I fully agree to publish it on Gels.

Reviewer 2 Report

The authors have addressed all my concerns. Minor textual editing may be helpful in improving the quality of the manuscript further.